# On-Clamp vs. Off-Clamp Robot-Assisted Partial Nephrectomy for cT2 Renal Tumors: Retrospective Propensity-Score-Matched Multicenter Outcome Analysis

**DOI:** 10.3390/cancers14184431

**Published:** 2022-09-13

**Authors:** Aldo Brassetti, Giovanni E. Cacciamani, Andrea Mari, Juan D. Garisto, Riccardo Bertolo, Chandru P. Sundaram, Ithaar Derweesh, Ahmet Bindayi, Prokar Dasgupta, James Porter, Alexander Mottrie, Luigi Schips, Koon Ho Rah, David Y. T. Chen, Chao Zhang, Kenneth Jacobsohn, Umberto Anceschi, Alfredo M. Bove, Manuela Costantini, Mariaconsiglia Ferriero, Riccardo Mastroianni, Leonardo Misuraca, Gabriele Tuderti, Alexander Kutikov, Wesley M. White, Stephen T. Ryan, Francesco Porpiglia, Jihad Kaouk, Andrea Minervini, Inderbir Gill, Riccardo Autorino, Giuseppe Simone

**Affiliations:** 1Department of Urology, IRCCS “Regina Elena” National Cancer Institute, 00144 Rome, Italy; 2USC Institute of Urology and Catherine, Joseph Aresty Department of Urology, Keck School of Medicine, Los Angeles, CA 90033, USA; 3Department of Experimental and Clinical Medicine, Oncologic Minimally Invasive Urology and Andrology Unit, Careggi Hospital, University of Florence, 50134 Florence, Italy; 4Department of Urology, Cleveland Clinic, Cleveland, OH 44195, USA; 5Division of Urology, San Carlo di Nancy Hospital, 00165 Rome, Italy; 6Department of Urology, Indiana University, Indianapolis, IN 47405, USA; 7Department of Urology, UCSD Health System, La Jolla, CA 92103, USA; 8MRC Centre for Transplantation, Guy’s Hospital, King’s College, London WC2R 2LS, UK; 9Swedish Urology Group, Seattle, WA 98104, USA; 10Department of Urology, OLV Hospital, 9300 Aalst, Belgium; 11Department of Urology, Annunziata Hospital, G. D’Annunzio University, 66100 Chieti, Italy; 12Urological Science Institute, Yonsei University College of Medicine, Seoul 03722, Korea; 13Division of Urologic Oncology, Fox Chase Cancer Center, Philadelphia, PA 19111, USA; 14Department of Urology, Changhai Hospital, Shanghai 200433, China; 15Department of Urology, Medical College Wisconsin, Milwaukee, WA 53226, USA; 16Department of Urology, University of Tennessee Medical Center, Knoxville, TN 37920, USA; 17Division of Urology, San Luigi Gonzaga Hospital, University of Turin, 10124 Orbassano, Italy; 18Division of Urology, Department of Surgery, Virginia Commonwealth University Health System, Richmond, VA 23298, USA

**Keywords:** partial nephrectomy, robot-assisted, on-clamp, hilar clamping, off-clamp, renal neoplasm, renal mass, clinical T2, outcomes

## Abstract

**Simple Summary:**

Robot-assisted partial nephrectomy is a viable option to treat tumors >7 cm. In this setting, however, the risk of postoperative renal function deterioration is higher because of the prolonged warm ischemia times (>20 min) that are required in two-thirds of cases, if an on-clamp approach is used. On the contrary, an off-clamp robotic partial nephrectomy is associated with shorter operation times and significantly improves the probabilities of achieving favorable perioperative outcomes.

**Abstract:**

We compared perioperative outcomes after on-clamp versus off-clamp robot-assisted partial nephrectomy (RAPN) for >7 cm renal masses. A multicenter dataset was queried for patients who had undergone RAPN for a cT2cN0cM0 kidney tumor from July 2007 to February 2022. The Trifecta achievement (negative surgical margins, no severe complications, and ≤ 30% postoperative estimated glomerular filtration rate (eGFR) reduction) was considered a surrogate of surgical quality. Overall, 316 cases were included in the analysis, and 58% achieved the Trifecta. A propensity-score-matched analysis generated two cohorts of 89 patients homogeneous for age, ASA score, preoperative eGFR, and RENAL score (all *p* > 0.21). Compared to the on-clamp approach, OT was significantly shorter in the off-clamp group (80 vs. 190 min; *p* < 0.001), the incidence of sRFD was lower (22% vs. 40%; *p* = 0.01), and the Trifecta rate higher (66% vs. 46%; *p* = 0.01). In a crude analysis, >20 min of hilar clamping was associated with a significantly higher risk of sRFD (OR: 2.30; 95%CI: 1.13–4.64; *p* = 0.02) and with reduced probabilities of achieving the Trifecta (OR: 0.46; 95%CI: 0.27–0.79; *p* = 0.004). Purely off-clamp RAPN seems to be a safe and viable option to treat cT2 renal masses and may outperform the on-clamp approach regarding perioperative surgical outcomes.

## 1. Introduction

Partial nephrectomy (PN) is the standard of care for cT1 renal masses [1], and growing evidence supports its role in selected cT2 cases as well [2,3]. In recent years, the robotic system has been broadly adopted for kidney surgery [4,5], and robot-assisted PN (RAPN) has become prevalent as it reduces complications and readmissions rates [6].

Hilar clamping limits intraoperative bleeding and allows a precise tumor resection; the temporary interruption of blood flow may cause ischemic injury, thus potentially undermining the intent of renal function (RF) preservation [7]. In the attempt to minimize ischemia time, the off-clamp approach has emerged as a viable option [8], although it is technically demanding, and the detrimental effect of blood flow suspension in patients with a contralateral healthy kidney is debated [7]. 

The aim of the present study was to compare the perioperative surgical outcomes of the on-clamp (on-RAPN) and off-clamp (off-RAPN) robotic PN for cT2 tumors in a large multi-institutional database.

## 2. Materials and Methods

Once an institutional review board approval or exemption was obtained from each of the 17 participating institutions, the purpose-built dataset was queried for patients with non-metastatic cT2 renal masses who had undergone RAPN within the study period (July 2007–February 2022). The following parameters were collected:Patients’ baseline characteristics (age, gender, race, body mass index (BMI), American Society of Anesthesiologists (ASA) score, solitary kidney status).Tumor characteristics (side, clinical size, stage [9], and surgical complexity graded according to the RENAL nephrometry score [10]).Perioperative variables (clamping technique, warm ischemia time (WIT), operation time (OT), length of hospital stay (LOS), postoperative complications (stratified according to the Clavien–Dindo (CD) system [11]; those ≥ grade III were defined as “major complications”. All the cases requiring intra-/postoperative blood transfusion and/or renal artery embolization were considered “major bleeding events” (MBE)).Pathology data (margin status, tumor size, stage [9], and histology [12]).Functional data (pre- and postoperative estimated glomerular filtration rate (eGFR) were calculated by the Modification of Diet in Renal Disease formula [13] and stratified according to the National Kidney Foundation (NKF) and its Kidney Disease Outcomes Quality Initiative [14]. According to the NKF and the US Food and Drug Administration, a significant renal function deterioration (sRFD) was defined as a >30% postoperative eGFR reduction [15]. Postoperative blood tests were performed before hospital discharge).

### 2.1. Study Objective

The aim of the present study was to compare perioperative surgical outcomes of the on-clamp and off-clamp approaches in patients who had undergone RAPN for cT2 renal masses. For this purpose, the achievement of the Trifecta (negative surgical margins, no postoperative major complications, and absence of sRFD) was used as a surrogate of surgical quality [16].

### 2.2. Statistical Analysis

The study population was split into two groups according to the hilar clamping technique (onRAPN vs. offRAPN). A 1:1 propensity score matching (PSM) analysis was used to minimize imbalances between the two cohorts for variables potentially affecting perioperative outcomes (age, ASA score, eGFR at baseline, RENAL score); the model was set to provide a standardized mean difference < 10% between covariates.

Frequencies and proportions were used to report categorical variables that were compared by means of the Chi-squared test. Continuous variables were presented as median and interquartile ranges (IQRs) and were compared using either the Mann–Whitney U test or Kruskal–Wallis one-way based on their normal or not-normal distribution, respectively (normality of the distribution of variables was tested by the Kolmogorov–Smirnov test). Predictors of Trifecta achievement were identified using the univariable logistic regression model; to reduce the risk of collinearity, the variables included in the definition of Trifecta were excluded from the multivariate analysis. Odds ratios and 95% confidence intervals (95%CIs) were reported. An alpha value of 5% was considered as a threshold for significance. Statistical analysis was performed using Statistical Package for Social Science 25.0 Software (SPSS Inc., Chicago, IL, USA). 

## 3. Results

Overall, 316 patients were included in the analysis, with a median age of 60 yrs (IQR: 51–67). The average BMI was 27.1 (IQR: 24.6–31.4), and 31% (n = 99) were obese. The median tumor size at final pathology was 80 mm (IQR: 64–95), and 37% (n = 117) of these masses were classified as RENAL score ≥10. A pT3a cancer was diagnosed in 17% (n = 44/258) of cases. The on-clamp approach was used in 211 (67%) cases overall; 140/211 (66%) required >20 min of warm ischemia (Table 1). The median OT was 180 min (IQR: 91.5/205), and it was significantly longer in the on-clamp group (190 min vs. 75 min; *p* < 0.001). Overall, the Trifecta rate was 58% (n = 182), and it was significantly lower in the on-clamp cohort (53% vs. 71%; *p* = 0.01) (Figure 1). Positive surgical margins (pSMs) were observed in 18 patients (6%); 42 (13%) experienced severe complications, and 97 (31%) were diagnosed with an sRFD. The latter finding was more common after on-clamp surgery (35% vs. 21%; *p* = 0.01) (Table 1) (Figure 1 and Figure 2).

The PSM analysis generated two cohorts of 89 patients homogeneous for age, ASA score, preoperative eGFR, and RENAL score (all *p* > 0.21) (Table 1). Compared to onRAPN, OT was significantly shorter in the offRAPN cohort (*p* < 0.001), the incidence of sRFD was lower (*p* = 0.01), and the Trifecta rate higher (*p* = 0.01). In a crude analysis, >20 min of hilar clamping was associated with a significantly higher risk of sRFD (OR: 2.30; 95%CI: 1.13–4.64; *p* = 0.02) and with reduced probabilities of achieving the Trifecta (OR: 0.46; 95%CI: 0.27–0.79; *p* = 0.004) (Table 2).

## 4. Discussion

Nephron-sparing surgery (NSS), whenever feasible, represents the standard of care for the treatment of cT1a renal masses [1] as it provides superior renal function preservation compared to RN [17]. However, the latter is still considered the gold-standard option for larger localized tumors, despite growing evidence supporting the role of PN in selected cases. In fact, a recent systematic review and meta-analysis assessed functional, oncologic, and perioperative outcomes of partial and RN in ≥ cT1b renal masses and highlighted that the former was associated with a lower decline in eGFR (WMD 8.6 mL/min; *p* < 0.001) and lower incidence of postoperative onset of renal failure (RR 0.36; *p* < 0.001), while tumor recurrence (OR 0.6; *p* < 0.001), cancer-specific mortality (OR 0.58; *p* = 0.001), and all-cause mortality (OR 0.67; *p* = 0.005) were less likely among patients undergoing NSS [18]. Only a few studies compared NSS to the “reference standard” RN in the specific setting of T2 tumors [19,20,21,22], and one accounted for the robot-assisted approach [23]. There were no significant differences in intraoperative (RAPN 6.9% vs. RN 5.3%; *p* = 0.478) and major postoperative complications (5.3% vs. 2.3%; *p* = 0.063). Additionally, 5-year overall survival (76.3% vs. 88.0%, *p* = 0.221) and disease-free survival (78.6% vs. 85.3%, *p* = 0.630) rates were comparable for pT2 renal cell carcinomas and freedom from de novo eGFR of <45 mL/min/1.73 m2 was 91.6% for RAPN vs. 68.9% for RN (*p* < 0.001) at the same timepoint [23].

In recent decades, thanks to the widespread utilization of robotic surgical platforms in the field of urology, confidence has exponentially grown, leading their utilization to perform more complex surgeries. Nonetheless, RAPN for tumors larger than 7 cm remains demanding, and only a few series have been published to date [3,24,25]. According to a large (n = 298), multicenter, retrospective study, approximately half of the treated patients achieved favorable surgical outcomes; the median postoperative eGFR decrease was 17.5%, and this deterioration of renal function remained substantially stable from discharge to the 12-month follow-up [3].

As said, a significant decline in function after NSS is observed in the treated kidney, and it is mainly due to the resected healthy parenchymal margin, ischemia–reperfusion damage at the time of hilar clamping, and “reconstructive injury” caused by renorrhaphy [26]. Recent studies provided evidence that both resection techniques (enucleation vs. enucleoresection) and WIT are independent predictors of postoperative acute renal failure [27]. As “every minute counts when the renal hilum is clamped during partial nephrectomy” [28], different techniques to minimize hypoperfusion were conceived, such as preoperative selective tumor embolization [29], super-selective clamping [30], early unclamping [31], and purely off-clamp PN [32].

In the present retrospective analysis, data on 316 patients who had undergone RAPN for cT2 renal masses were included, and 37% (n = 117) presented with highly complex tumors (RENAL ≥ 10). We reported a 6% pSM rate, consistent with that of a recent large series of minimally invasive PNs from the United States National Cancer Database [33]. No difference was observed between on-clamp and off-clamp cohorts (6% vs. 5%; *p* = 0.61), and this finding was confirmed by PSM analysis. Additionally, the incidence of MBEs was comparable in the two groups. These observations are of great interest as hilar clamping is traditionally performed to provide a relatively bloodless surgical field that theoretically ensures precise tumor resection while limiting intraoperative blood loss. The Trifecta rate was 58% overall, significantly lower than that in reports from the RAPN series on small renal masses [16,34]. However, this finding is not unexpected because increasing tumor complexity is an independent predictor of surgical quality [16]. Our results align with those observed in patients who have undergone NSS for complex, endophytic renal tumors [35].

In the present series, 105 patients underwent purely off-clamp RAPN. Although tumor size and surgical complexity were comparable in the two groups, OT time was significantly shorter in this cohort compared to on-clamp NSS patients (75 min vs. 190 min; *p* < 0.001), as renal hilum dissection was not routinely required. This observation is not surprising, and comparable findings are described in reports from the RAPN series on cT1 renal masses [36]. An on-clamp approach was used in two-thirds of cases (n = 211), and 140/211 required >20 min of warm ischemia. Prolonged clamping times are not usually needed when small renal tumors are resected [16], but when facing large masses, average WIT may exceed 30 min if a minimally invasive approach is used [25]. The impact of vascular clamping on postoperative renal function is still debated as most patients present with two functioning kidneys and show eGFR values within the normal range at baseline. Conversely, the advantages of a purely off-clamp approach are obvious in patient candidates for imperative NSS [36]. Thompson et al. provided evidence that every minute of warm ischemia significantly affects postoperative renal function [28] and pointed out that >20 min of vascular clamping during NSS is associated with a higher incidence of acute and chronic renal failure compared to patients with no ischemia [37]. Although a high rate of sRFD (31%) was observed in the present study population overall, the share was significantly higher in the on-clamp cohort compared to the off-clamp PN cohort (35% vs. 21%; *p* = 0.01), and this finding was confirmed by PSM (40% vs. 22%; *p* = 0.01). Moreover, according to our logistic regression analyses, exceeding 20 min of WIT significantly increased the risk of sRFD (OR: 2.30; 95%CI: 1.13–4.64; *p* = 0.02) and reduced the likelihood of achieving the Trifecta (OR: 0.46; 95%CI: 0.27–0.79; *p* = 0.004).

The present study was not devoid of limitations. First, the retrospective design accounts for some inherent biases, including patient selection. Moreover, although the use of eGFR is a practical, viable option, nuclear scans should be preferred to assess renal function. In addition, since this was a multicenter study, different surgical techniques were used during RAPN (complete hilum clamping vs. artery-only control; renorrhaphy vs. sutureless…), and some of these might have determined different functional outcomes across the centers. Another major limitation was the lack of data concerning intraoperative complications, which may have affected OT. Furthermore, as we aimed at comparing surgical outcomes after off-clamp vs. on-clamp RAPN, follow-up data were not collected, and long-term functional and oncologic outcomes are missing. In this regard, Zhang et al. have already provided evidence that even in patients with >2 fold increased postoperative creatinine, up to 90% of recovery can be expected [38]. Additionally, although the RENAL score (and mainly the “Radius” and “exophytic/endophytic properties”) was found to be an independent predictor of postoperative glomerular filtration rate decline [39], data required to calculate the tumor contact surface area (which was shown to be a good proxy of both excisional volume loss and renorrhaphy-related renal damage) [40] were missing, so this preoperative variable could not be included in the PSM analysis, possibly leading to selection biases. Last, all the participating institutions are high-volume referral centers, which limits our findings’ generalizability to the entire urological community.

Notwithstanding these limitations, ours is the first study that compares surgical outcomes of on-clamp and off-clamp RAPN in the setting of cT2 renal masses, and our cohort is the largest available concerning the robotic nephron-sparing approach to large kidney tumors.

## 5. Conclusions

In experienced hands, purely off-clamp RAPN is a safe and viable option for cT2 renal masses. It offers the potential advantages of a shorter operative time and lower decline in renal function in the immediate postoperative period.

## Figures and Tables

**Figure 1 cancers-14-04431-f001:**
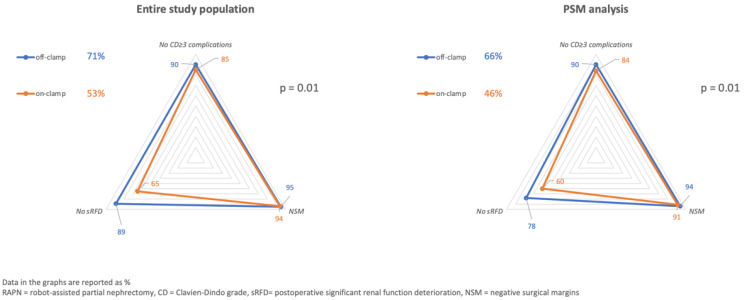
Trifecta rates, according to the clamping technique.

**Figure 2 cancers-14-04431-f002:**
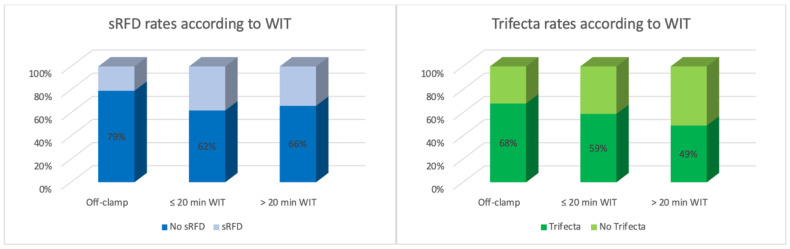
Severe renal function deterioration (sRFD) and Trifecta rates according to WIT.

**Table 1 cancers-14-04431-t001:** Patients’ baseline characteristics and outcomes after robot-assisted partial nephrectomy, according to the clamping technique.

	Overall (n = 316)	On-Clamp (n = 211)	Off-Clamp (n = 105)	*p*	PSM On-Clamp (n = 89)	PSM Off-Clamp (n = 89)	*p*
Age, years	60 (51/67)	60 (52/67)	60 (46/67)	0.21	59 (53/67)	60 (48/67)	0.58
Male gender, n (%)	202 (64%)	145 (69%)	57 (54%)	0.01	63 (71%)	50 (56%)	0.04
BMI	27.1 (24.6/31.4)	28.7 (25.3/32.5)	25.8 (23.9/29.3)	<0.001	26.6 (23.9/30.4)	26 (24.2/30)	0.05
Preop-eGFR, mL/min/1.73 m^2^	79 (64/95)	78 (61/94.6)	80 (66.1/96)	0.43	77 (61.2/92.8)	80.8 (66.1/95.2)	0.49
ASA score ≥ 3, n (%)	112 (35%)	79 (37%)	33 (31%)	0.29	37 (42%)	29 (33%)	0.21
Solitary kidney, n	13 (4%)	9 (4%)	4 (4%)	0.848	3 (3%)	4 (4%)	0.70
Clinical tumor size, mm	80 (73/86)	77 (73/85)	80 (75/90)	<0.001	80 (75/89)	80 (75/85)	0.51
RENAL score ≥ 10, n (%)	117 (37%)	66 (31%)	51 (49%)	0.005	32 (36%)	38 (43%)	0.09
OT, min	180 (91.5/205)	190 (170/225)	75 (70/110)	<0.001	190 (161.5/230)	80 (70/112.5)	<0.001
LOS, d	4 (3/5)	4 (3/5)	4 (3/5)	0.51	4 (3/5)	4 (3/5)	0.12
Benign histology, n (%)	58 (18%)	36 (17%)	22 (21%)	0.40	14 (16%)	16 (18%)	0.69
Postop-eGFR, mL/min/1.73 m^2^	68 (50.2/83.7)	64.5 (49.2/80.7)	73 (57/90)	0.02	66 (48.8/76.5)	70 (57/84.9)	0.14
% eGFR loss *	12 (2/26)	13 (2/31)	10 (2/20)	0.04	16 (0/31)	10 (2/20)	0.09
Major bleeding events, n	41 (13%)	24 (11%)	17 (16%)	0.23	13 (15%)	13 (15%)	1.00
Trifecta pSM, n (%) Complications CD ≥ 3, n (%) sRFD, n (%)	182 (58%) 18 (6%) 42 (13%) 97 (31%)	111 (53%) 13 (6%) 31 (15%) 75 (35%)	68 (71%) 5 (5%) 11 (10%) 22 (21%)	0.01 0.61 0.30 0.01	41 (46%) 8 (9%) 14 (16%) 36 (40%)	59 (66%) 5 (6%) 9 (10%) 20 (22%)	0.01 0.39 0.26 0.01

Data are reported as median (IQR). * baseline vs. discharge. PSM = propensity-score-matched analysis, BMI = body mass index, Preop-eGFR = preoperative estimated glomerular filtration rate, ASA = American Society of Anesthesiologists, WIT = warm ischemia time, OT = operative time, CKD = chronic kidney disease, LOS = length of stay, Postop-eGFR = postoperative estimated glomerular filtration rate, ∆eGFR = postoperative reduction of estimated glomerular filtration rate, pSM = positive surgical margins, CD = Clavien–Dindo complication scale, sRFD = significant renal function deterioration.

**Table 2 cancers-14-04431-t002:** Logistic regression analyses to identify predictors of severe renal function deterioration (sRFD) (**A**) and Trifecta achievement (**B**) probabilities.

**A**	**Significant (<30%) Renal Function Deterioration**							
	**Univariable Analysis**				**Multivariable Analysis**			
	**OR**	**95% CI**		***p*-Value**	**OR**	**95% CI**		***p*-Value**
		**Lower**	**Higher**			**Lower**	**Higher**	
Age	1.02	0.99	1.04	0.19	-	-	-	-
Male gender	1.32	0.68	2.58	0.41	-	-	-	-
BMI	1.03	0.97	1.09	0.39	-	-	-	-
ASA score ≥ 3	1.28	0.67	2.45	0.45	-	-	-	-
Solitary kidney	0.87	0.16	4.61	0.87	-	-	-	-
Clinical tumor size	0.99	0.96	1.01	0.27	-	-	-	-
RENAL score ≥ 10	0.99	0.52	1.90	0.99	-	-	-	-
Preop-eGFR	1.01	0.99	1.03	0.06	-	-	-	-
WIT, min	1.02	1.00	1.04	0.02	-	-	-	-
Off-clamp	ref	-	-	0.38				
≤20 min of clamping	2.46	0.95	6.38	0.06				
>20 min of clamping	2.30	1.13	4.64	0.02				
**B**	**Trifecta Achievement**							
	**Univariable Analysis**				**Multivariable Analysis**			
	**OR**	**95% CI**		** *p* ** **-Value**	**OR**	**95% CI**		** *p* ** **-Value**
		**Lower**	**Higher**			**Lower**	**Higher**	
Age	0.53	0.97	1.02	0.99	-	-	-	-
Male gender	0.86	0.47	1.60	0.64	-	-	-	-
BMI	1.008	0.966	1.051	0.710	-	-	-	-
ASA score ≥ 3	0.74	0.40	1.37	0.34	-	-	-	-
Solitary kidney	0.30	0.06	1.58	0.15	-	-	-	-
Clinical tumor size	1.01	0.98	1.03	0.62	-	-	-	-
RENAL score ≥ 10	1.29	0.70	2.38	0.409	-	-	-	-
Preop-eGFR	0.99	0.97	1.01	0.06	-	-	-	-
WIT, min	0.98	0.96	0.99	0.02	-	-	-	-
Off-clamp	ref	-	-	0.38				
≤20 min of clamping	0.69	0.37	1.29	0.25				
>20 min of clamping	0.46	0.27	0.79	0.004				

## Data Availability

Data supporting reported results are deposited at https://gbox.garr.it/ (accessed on 1 July 2022).

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
