# Peer review of "On-Clamp vs. Off-Clamp Robot-Assisted Partial Nephrectomy for cT2 Renal Tumors: Retrospective Propensity-Score-Matched Multicenter Outcome Analysis"

_cancers, 2022, doi:10.3390/cancers14184431_

Round 1

Reviewer 1 Report

The authors compared perioperative outcome after on-clamp versus off-clamp RAPN to cT2 renal mass. The background is aligned by implementing PSM.

The results show that the off-clamp group is an effective surgical method in terms of OT, the incidence of sRFD (>30% eGFR reduction before hospital discharge) and achievement rate of trifecta.

I have a few questions

â‘     The degree of decrease in eGFR is shown only by sRFD, but what was the specific value?

â‘¡    The discussion also stated that "eGFR that decreased immediately after surgery often improves", but how was the long-term transition of eGFR in this study?

â‘¢     I feel that the ratio of Benign histology is high in this study, but how is it compared to other reports and how was the Upstage ratio to cT3a?

Author Response

Reviewer #1

The authors compared perioperative outcome after on-clamp versus off-clamp RAPN to cT2 renal mass. The background is aligned by implementing PSM.

The results show that the off-clamp group is an effective surgical method in terms of OT, the incidence of sRFD (>30% eGFR reduction before hospital discharge) and achievement rate of trifecta.

We thank the reviewer for agreeing to review our paper.

I have a few questions:

  1. The degree of decrease in eGFR is shown only by sRFD, but what was the specific value?

We thank the reviewer for the question.

Actually, data concerning renal function before and after RAPN are already reported in Table 1.

To better clarify this relevant finding, as suggested by the reviewer, the percentages of eGFR loss were calculated and showed in Table 1 (page 5).

Overall, an average 12% (IQR: 2-26) eGFR reduction was observed after RAPN in the entire study population: this loss was significantly higher in the on-clamp group (13% vs 10%; p=0.047), compared to off-clamp.

After PSM analysis a difference was still observed between the two cohorts (16% vs 10%; p=0.089) but it did not reach statistical significance.

  1. The discussion also stated that "eGFR that decreased immediately after surgery often improves", but how was the long-term transition of eGFR in this study?

We thank the reviewer for his/her question.

As reported in the Materials and methods section (page 3, lines 97-98: “Post-operative blood tests were performed before hospital discharge”) and further acknowledged in the Discussion section (page 9, lines 237-238: “[…] follow-up data were not collected, and long-term functional […] outcomes are missing”) we only collected perioperative creatinine values and compared eGFR values before and right after surgery (before hospital discharge).

In fact, the aim of the present study was to compare the trifecta rate of off-clamp vs on-clamp RAPN in the specific setting of large renal masses (> 7 cm). Indeed, the assessment of mid-term and long-term functional outcomes in these cohort could be the topic of further investigations.

Up to today, we rely on the recently published findings of the ROSULA collaborative group: a 15% eGFR reduction was observed after RAPN for cT2 renal masses but it substantially recovered at 12 months follow-up[1]. However, considering than only 2.7% of the included patients presented with a solitary kidney, such recovery could be at least partially due to the compensation of the contralateral healthy organ and long term eGFR values do not provide info concerning the actual damage suffered by the treated kidney.

  1. I feel that the ratio of Benign histology is high in this study, but how is it compared to other reports and how was the Upstage ratio to cT3a?

We thank the reviewer for the comment.

The 18% rate of benign tumors that we observed is in line with the findings of Bertolo et al in their cohort of 298 cT2 renal masses treated with robotic partial nephrectomy[1]. Also Klett and colleagues reported a 12% share of benign lesions in their series of 451 patients undergone surgery for large renal masses; interestingly, such rate was significantly higher (31% vs 9%; p <0.001) in the nephron-sparing subgroup[2].

According to our data, 17% (n=44/258) of patients were diagnosed with a pT3a cancer at final pathology. Also Klett et al observed a comparable upstaging rate (18%) in their nephron-sparing subgroup[2]. On the contrary, according to data from the ROSULA collaborative group, the share of upstaged tumors after partial nephrectomy for large renal masses can be as high as 38%[1].  

Indeed, a reduced share of pT3a cancer may explain the limited rate of positive surgical margins observed in our study (6%), while it was only slightly higher (8%) in the series from the ROSULA group[1].

As suggested by the reviewer, the overall upstaging rate was reported in the manuscript:

  • Page 3, lines 128-129: “A pT3a cancer was diagnosed in 17% (n=44/258) of cases.”

Reviewer 2 Report

Comment to the author

This report describing On-clamp vs. off-clamp robot-assisted partial nephrectomy for 2 cT2 renal tumors: retrospective propensity score matched multi-3 center outcome analysis is well written. However, I noticed some problems in this article. Please carefully consider the comments as below.

Major

1) In this study, a retrospective comparison between the on-clamp and off-clamp group was performed. Furthermore, propensity score matching (PSM) was performed based on age, ASA score, baseline eGFR, and RENAL score. However, I feel that this PSM is insufficient to eliminate selection bias between the two groups. In particular, the operative time is too different between the two groups. The authors described the reason at discussion part, as renal hilum dissection was not routinely required, but it is difficult to imagine that omission of renal hilum dissection would result in a reduction of nearly 2 hours. There must be factors in the on-clamp group that make the surgery more difficult, such as hilar tumor, high PADUA score, high Mayo Adhesive Probability (MAP) score. If such factors are present, on-clamp RAPN would be selected. The authors should show these parameters such as the rate of hilar tumor cases, mean or median PADUA score, and mean or median MAP score. If necessary, these parameters should be used to perform PSM.

2) The volume of nephron loss, as well as the ischemia and re-perfusion injury, is the significant factor in the postoperative decline in renal function. Many articles have stated that the size of the contact surface area affects postoperative renal function. However, there appears to be no description of this. It should be mentioned because there seems to be a significant difference between the two groups.

3) The decrease in renal function due to inhibition is reversible, and the time of inhibition and the method of inhibition do not seem to have much effect on long-term renal function. Therefore, the authors should describe when the creatinine value measured was used to assess postoperative renal function.

4) In which cases with cT2 renal masses would off-clamp RAPN be especially useful? The authors should also mention about it.

Author Response

Reviewer #2

This report describing “On-clamp vs. off-clamp robot-assisted partial nephrectomy for 2 cT2 renal tumors: retrospective propensity score matched multi-3 center outcome analysis” is well written.

We thank the reviewer for agreeing to review our paper and for his/her comments.

However, I noticed some problems in this article. Please carefully consider the comments as below:

  • In this study, a retrospective comparison between the on-clamp and off-clamp group was performed. Furthermore, propensity score matching (PSM) was performed based on age, ASA score, baseline eGFR, and RENAL score. However, I feel that this PSM is insufficient to eliminate selection bias between the two groups. In particular, the operative time is too different between the two groups. The authors described the reason at discussion part, “as renal hilum dissection was not routinely required”, but it is difficult to imagine that omission of renal hilum dissection would result in a reduction of nearly 2 hours. There must be factors in the on-clamp group that make the surgery more difficult, such as hilar tumor, high PADUA score, high Mayo Adhesive Probability (MAP) score. If such factors are present, on-clamp RAPN would be selected. The authors should show these parameters such as the rate of hilar tumor cases, mean or median PADUA score, and mean or median MAP score. If necessary, these parameters should be used to perform PSM.

We thank the reviewer for the comment.

In order to mitigate possible selection biases between the on-clamp and the off-clamp groups, a propensity score matching analysis was performed, and all the available preoperative variables that could affect surgical and functional outcomes (age, comorbidities, renal function at baseline, and tumor surgical complexity) were considered. As tumor complexity has a remarkable impact on operation time, complication rate and postoperative renal function, it was defined according to the standardized and widely accepted RENAL score [10.1016/j.eururo.2011.03.029] and the significant difference initially observed between the two groups (49% of RENAL ≥10 tumors in the off-clamp cohort vs 31% in the on-clamp cohort; p = 0.005) was minimized through PSM analysis (43% vs 36%; p= 0.09).

Although the reviewer suggested using PADUA and MAP scores to better account for tumor location and surface of contact, there is no grounded evidence to support the superiority of these nephrometry scores compared to RENAL[3–6]. In fact, a recent systematic review and meta-analysis concluded that both RENAL and PADUA represent the best tools to report tumor complexity and predict treatment morbidity; despite other scores being promising, their role in forecasting specific outcomes does not outperform the first-generation ones[7].

The reviewer focused his/her attention on operation time, which was statistically longer in the on-clamp group both before (190 min vs 75 min; p <0.001) and after PSM analysis (190 min vs 80 min; p <0.001), compared to off-clamp. Considering that propensity score matching minimized imbalances between the two cohorts in terms of tumor complexity, we are persuaded that such difference is due to the various surgical techniques used to treat the patients in the two groups. In particular, three quarter of the off-clamp cases (79/105; 75%) included in the present study were performed in one single referral center (“Regina Elena” National Cancer Institute), where this surgical approach was pioneered and it is nowadays used in all the nephron sparing cases, regardless tumor complexity. According to the surgical technique popularized by Simone et al[8], hilar dissection is routinely avoided, kidney mobilization is minimized and renal tumor is directly identified and excised, thus allowing for reduced OT also in case of purely hilar[9] and totally endophytic[10] renal masses.

A recent systematic review on off-clamp partial nephrectomies supports our hypothesis: among the 13 included studies, the observed mean OT was always > 120 minutes except for the 3 papers published by Simone and colleagues that reported halved operation times[11].

  • The volume of nephron loss, as well as the ischemia and re-perfusion injury, is the significant factor in the postoperative decline in renal function. Many articles have stated that the size of the contact surface area affects postoperative renal function. However, there appears to be no description of this. It should be mentioned because there seems to be a significant difference between the two groups.

We thank the reviewer for his/her comment.

We totally agree with him/her that nephron loss after partial nephrectomy may arise from prolonged ischemia time, surgical excision of healthy parenchyma and iatrogenic injury from reconstruction[12]. The extent of the tumor contact surface area (CSA) is thought to be a good proxy of both excisional volume loss and renorrhaphy-related renal damage, and there is grounded evidence that it predicts postoperative ipsilateral renal function decline[13–15].

Unfortunately, considering the retrospective nature of the present study, CSA could not be calculated in all the 316 included patient, and it was not available for PSM analysis.

However, the RENAL score was available for all the cases included in the present study and it was used at PSM analysis, together with other preoperative variables, to construct two cohorts of patients with comparable baseline characteristics. After matching, clinical tumor size was similar in the two groups.

Interestingly, also this widely adopted nephrometry score is considered an independent predictor of renal function decline after nephron-sparing surgery and, with this regard, it seems not inferior to CSA[16].

Concluding, as per reviewer’s suggestion, we acknowledge the limitation as follows:

  • Page 9, lines 240-246: “Additionally, despite also RENAL score (and mainly “Radius” and “exophytic/endophytic properties”) was found to be an independent predictor of postoperative glomerular filtration rate decline[16], data required to calculate the tumor contact surface area (which was shown to be a good proxy of both excisional volume loss and renorrhaphy-related renal damage)[17] were missing so that this preoperative variable could not be included in the PSM analysis, possibly leading to selection biases.”
  • The decrease in renal function due to inhibition is reversible, and the time of inhibition and the method of inhibition do not seem to have much effect on long-term renal function. Therefore, the authors should describe when the creatinine value measured was used to assess postoperative renal function.

We thank the reviewer for the comment.

We well know that the impact of vascular clamping on post-operative renal function is still debated as most patients present with two functioning kidneys and show eGFR values within the normal range at baseline. Conversely, the advantages of a purely off-clamp approach are obvious in patients candidate for imperative nephron sparing surgery[18]. Thompson et al. provided evidence that every minute of warm ischemia significantly affects post-operative renal function[19] and pointed out that > 20 minutes of vascular clamping during partial nephrectomy are associated with a higher incidence of acute and chronic renal failure compared to patients with no ischemia[20].

According to our results, a high rate of sRFD (31%) was observed: the share was significantly higher in the on-clamp cohort compared to off-clamp (35% vs. 21%; p=0.01), and this finding was confirmed at PSM analysis (40% vs. 22%; p=0.01). Interestingly, exceeding 20 min of WIT significantly increased the risk sRFD (OR: 2.30; 95%CI: 1.13-4.64; p=0.02) at logistic regression analysis.

Assessing long-term renal function was beyond the scope of the present paper and post-operative eGFR was calculated according to creatinine values obtained at blood tests performed before hospital discharge, as already clarified in the Materials and Methods section (page 2-3, lines 92-98): “pre- and post-operative estimated glomerular filtration rate [eGFR] were calculated by the Modification of Diet in Renal Disease formula […]. Post-operative blood tests were performed before hospital discharge”.

The same limitation was also disclosed at the end of the discussion section (page 9, lines 236-240): “[…]as we aimed at comparing surgical outcomes after off-clamp vs. on-clamp RAPN, follow-up data were not collected, and long-term functional and oncologic outcomes are missing. With this regard, Zhang et al. already provided evidence that even in patients with a > 2 folds increased postoperative creatinine, up to 90% of recovery can be expected[21].”

4) In which cases with cT2 renal masses would off-clamp RAPN be especially useful? The authors should also mention about it.

We thank the reviewer for the comment.

The present study was not conceived to address this question and we have no data to identify the sub-population of patients best suitable for an off-clamp RAPN. Actually, also role of nephron-sparing surgery in cT2 cancer in the “elective setting” is still debated and international guidelines only recommend partial nephrectomy in solitary kidney or chronic kidney disease, if technically feasible.

In the present study, no standardized criteria were shared by the 17 participating institutions in order to select the best treatment option for each patient: the choice between the on-clamp and the off-clamp approaches was only based on surgeon’s confidence with a specific technique, instead. As a result, 75% of the off-clamp cases were performed in one single institution where this approach was pioneered and has been used for all the nephron sparing surgeries in the last two decades.

For these reasons, according to our findings, we could only state that (page 9, lines 253-255) “[…] purely off-clamp RAPN is a safe and viable option for cT2 renal masses” and, in experienced hands, “it offers potential advantages of shorter operative time and lower decline of renal function in the immediate postoperative period”.

References:

  1. Bertolo, R.; Autorino, R.; Simone, G.; Derweesh, I.; Garisto, J.D.; Minervini, A.; Eun, D.; Perdona, S.; Porter, J.; Rha, K.H.; et al. Outcomes of Robot-Assisted Partial Nephrectomy for Clinical T2 Renal Tumors: A Multicenter Analysis (ROSULA Collaborative Group). Eur Urol 2018, 74, 226–232, doi:10.1016/j.eururo.2018.05.004.
  2. Klett, D.E.; Tsivian, M.; Packiam, V.T.; Lohse, C.M.; Ahmed, M.E.; Potretzke, T.A.; Gopalakrishna, A.; Boorjian, S.A.; Thompson, R.H.; Leibovich, B.C.; et al. Partial versus Radical Nephrectomy in Clinical T2 Renal Masses. Int J Urol 2021, 28, 1149–1154, doi:10.1111/iju.14664.
  3. Karamık, K.; AktaÅŸ, Y.; Erdemir, A.G.; Ä°slamoÄŸlu, E.; Ölçücü, M.T.; Özsoy, Ç.; SavaÅŸ, M.; AteÅŸ, M. Predicting Strict Trifecta Outcomes after Robot-Assisted Partial Nephrectomy: Comparison of RENAL, PADUA, and C-Index Scores. J Kidney Cancer VHL 2021, 8, 1–12, doi:10.15586/jkcvhl.v8i4.183.
  4. Sharma, A.P.; Mavuduru, R.S.; Bora, G.S.; Devana, S.K.; Palani, K.; Lal, A.; Kakkar, N.; Singh, S.K.; Mandal, A.K. Comparison of RENAL, PADUA, and C-Index Scoring Systems in Predicting Perioperative Outcomes after Nephron Sparing Surgery. Indian J Urol 2018, 34, 51–55, doi:10.4103/iju.IJU_247_17.
  5. Wang, Y.-D.; Huang, C.-P.; Chang, C.-H.; Wu, H.-C.; Yang, C.-R.; Wang, Y.-P.; Hsieh, P.-F. The Role of RENAL, PADUA, C-Index, CSA Nephrometry Systems in Predicting Ipsilateral Renal Function after Partial Nephrectomy. BMC Urol 2019, 19, 72, doi:10.1186/s12894-019-0504-2.
  6. Schiavina, R.; Novara, G.; Borghesi, M.; Ficarra, V.; Ahlawat, R.; Moon, D.A.; Porpiglia, F.; Challacombe, B.J.; Dasgupta, P.; Brunocilla, E.; et al. PADUA and R.E.N.A.L. Nephrometry Scores Correlate with Perioperative Outcomes of Robot-Assisted Partial Nephrectomy: Analysis of the Vattikuti Global Quality Initiative in Robotic Urologic Surgery (GQI-RUS) Database. BJU Int 2017, 119, 456–463, doi:10.1111/bju.13628.
  7. Veccia, A.; Antonelli, A.; Uzzo, R.G.; Novara, G.; Kutikov, A.; Ficarra, V.; Simeone, C.; Mirone, V.; Hampton, L.J.; Derweesh, I.; et al. Predictive Value of Nephrometry Scores in Nephron-Sparing Surgery: A Systematic Review and Meta-Analysis. Eur Urol Focus 2020, 6, 490–504, doi:10.1016/j.euf.2019.11.004.
  8. Simone, G.; Misuraca, L.; Tuderti, G.; Minisola, F.; Ferriero, M.; Romeo, G.; Costantini, M.; Al-Rawashdah, S.F.; Guaglianone, S.; Gallucci, M. Purely Off-Clamp Robotic Partial Nephrectomy: Preliminary 3-Year Oncological and Functional Outcomes. Int J Urol 2018, 25, 606–614, doi:10.1111/iju.13580.
  9. Ferriero, M.; Brassetti, A.; Mastroianni, R.; Costantini, M.; Tuderti, G.; Anceschi, U.; Bove, A.M.; Misuraca, L.; Guaglianone, S.; Gallucci, M.; et al. Off-Clamp Robot-Assisted Partial Nephrectomy for Purely Hilar Tumors: Technique, Perioperative, Oncologic and Functional Outcomes from a Single Center Series. Eur J Surg Oncol 2022, S0748-7983(22)00066-X, doi:10.1016/j.ejso.2022.01.024.
  10. Tuderti, G.; Brassetti, A.; Mastroianni, R.; Misuraca, L.; Bove, A.; Anceschi, U.; Ferriero, M.; Guaglianone, S.; Gallucci, M.; Simone, G. Expanding the Limits of Nephron-Sparing Surgery: Surgical Technique and Mid-Term Outcomes of Purely off-Clamp Robotic Partial Nephrectomy for Totally Endophytic Renal Tumors. Int J Urol 2022, 29, 282–288, doi:10.1111/iju.14763.
  11. Simone, G.; Gill, I.S.; Mottrie, A.; Kutikov, A.; Patard, J.-J.; Alcaraz, A.; Rogers, C.G. Indications, Techniques, Outcomes, and Limitations for Minimally Ischemic and off-Clamp Partial Nephrectomy: A Systematic Review of the Literature. Eur Urol 2015, 68, 632–640, doi:10.1016/j.eururo.2015.04.020.
  12. Mir, M.C.; Ercole, C.; Takagi, T.; Zhang, Z.; Velet, L.; Remer, E.M.; Demirjian, S.; Campbell, S.C. Decline in Renal Function after Partial Nephrectomy: Etiology and Prevention. J Urol 2015, 193, 1889–1898, doi:10.1016/j.juro.2015.01.093.
  13. Kahn, A.E.; Shumate, A.M.; Galler, I.J.; Ball, C.T.; Thiel, D.D. Contact Surface Area and Its Association with Outcomes in Robotic-Assisted Partial Nephrectomy. Int J Med Robot 2020, 16, e2069, doi:10.1002/rcs.2069.
  14. Takagi, T.; Yoshida, K.; Kondo, T.; Kobayashi, H.; Iizuka, J.; Okumi, M.; Ishida, H.; Tanabe, K. Association between Tumor Contact Surface Area and Parenchymal Volume Change in Robot-Assisted Laparoscopic Partial Nephrectomy Carried out Using the Enucleation Technique. Int J Urol 2019, 26, 745–751, doi:10.1111/iju.14004.
  15. Ficarra, V.; Crestani, A.; Bertolo, R.; Antonelli, A.; Longo, N.; Minervini, A.; Novara, G.; Simeone, C.; Carini, M.; Mirone, V.; et al. Tumour Contact Surface Area as a Predictor of Postoperative Complications and Renal Function in Patients Undergoing Partial Nephrectomy for Renal Tumours. BJU Int 2019, 123, 639–645, doi:10.1111/bju.14567.
  16. Gupta, R.; Tori, M.; Babitz, S.K.; Tobert, C.M.; Anema, J.G.; Noyes, S.L.; Lane, B.R. Comparison of RENAL, PADUA, CSA, and PAVP Nephrometry Scores in Predicting Functional Outcomes After Partial Nephrectomy. Urology 2019, 124, 160–167, doi:10.1016/j.urology.2018.03.055.
  17. Ficarra, V.; Crestani, A.; Bertolo, R.; Antonelli, A.; Longo, N.; Minervini, A.; Novara, G.; Simeone, C.; Carini, M.; Mirone, V.; et al. Tumour Contact Surface Area as a Predictor of Postoperative Complications and Renal Function in Patients Undergoing Partial Nephrectomy for Renal Tumours. BJU Int 2019, 123, 639–645, doi:10.1111/bju.14567.
  18. ANCESCHI, U.; BRASSETTI, A.; BERTOLO, R.; TUDERTI, G.; FERRIERO, M.C.; MASTROIANNI, R.; FLAMMIA, R.S.; COSTANTINI, M.; KAOUK, J.; LEONARDO, C.; et al. On-Clamp versus Purely off-Clamp Robot-Assisted Partial Nephrectomy in Solitary Kidneys: Comparison of Perioperative Outcomes and Chronic Kidney Disease Progression at Two High- Volume Centers. Minerva urologica e nefrologica = The Italian journal of urology and nephrology 2020, doi:10.23736/S0393-2249.20.03795-9.
  19. Thompson, R.H.; Lane, B.R.; Lohse, C.M.; Leibovich, B.C.; Fergany, A.; Frank, I.; Gill, I.S.; Blute, M.L.; Campbell, S.C. Every Minute Counts When the Renal Hilum Is Clamped during Partial Nephrectomy. Eur Urol 2010, 58, 340–345, doi:10.1016/j.eururo.2010.05.047.
  20. Thompson, R.H.; Frank, I.; Lohse, C.M.; Saad, I.R.; Fergany, A.; Zincke, H.; Leibovich, B.C.; Blute, M.L.; Novick, A.C. The Impact of Ischemia Time during Open Nephron Sparing Surgery on Solitary Kidneys: A Multi-Institutional Study. J Urol 2007, 177, 471–476, doi:10.1016/j.juro.2006.09.036.
  21. Zhang, Z.; Zhao, J.; Dong, W.; Remer, E.; Li, J.; Demirjian, S.; Zabell, J.; Campbell, S.C. Acute Kidney Injury after Partial Nephrectomy: Role of Parenchymal Mass Reduction and Ischemia and Impact on Subsequent Functional Recovery. Eur Urol 2016, 69, 745–752, doi:10.1016/j.eururo.2015.10.023.

Round 2

Reviewer 2 Report

We would like to thank the authors for their corrections.

We have no particular comments on the revised paper.